# Gibberellin Promotes Bolting and Flowering via the Floral Integrators *RsFT* and *RsSOC1-1* under Marginal Vernalization in Radish

**DOI:** 10.3390/plants9050594

**Published:** 2020-05-07

**Authors:** Haemyeong Jung, Seung Hee Jo, Won Yong Jung, Hyun Ji Park, Areum Lee, Jae Sun Moon, So Yoon Seong, Ju-Kon Kim, Youn-Sung Kim, Hye Sun Cho

**Affiliations:** 1Plant Systems Engineering Research Center, Korea Research Institute of Bioscience and Biotechnology, Daejeon 34141, Korea; hmjung@kribb.re.kr (H.J.); chohee0720@kribb.re.kr (S.H.J.); jwy95@kribb.re.kr (W.Y.J.); hjpark@kribb.re.kr (H.J.P.); lar1027@kribb.re.kr (A.L.); jsmoon@kribb.re.kr (J.S.M.); 2Department of Biosystems and Bioengineering, KRIBB School of Biotechnology, Korea University of Science and Technology, Daejeon 34113, Korea; 3Crop Biotechnology Institute/GreenBio Science and Technology, Seoul National University, Pyeongchang 25354, Korea; syseong7@snu.ac.kr (S.Y.S.); jukon@snu.ac.kr (J.-K.K.); 4Graduate School of International Agricultural Technology, Seoul National University, Pyeongchang 25354, Korea; 5Department of Biotechnology, NongWoo Bio, Anseong 17558, Korea

**Keywords:** bolting, flowering time gene, gibberellin, radish (*Raphanus sativus* L.), RNA sequencing, *RsFT*, *RsSOC1*, vernalization

## Abstract

Gibberellic acid (GA) is one of the factors that promotes flowering in radish (*Raphanus Sativus* L.), although the mechanism mediating GA activation of flowering has not been determined. To identify this mechanism in radish, we compared the effects of GA treatment on late-flowering (NH-JS1) and early-flowering (NH-JS2) radish lines. GA treatment promoted flowering in both lines, but not without vernalization. NH-JS2 plants displayed greater bolting and flowering pathway responses to GA treatment than NH-JS1. This variation was not due to differences in GA sensitivity in the two lines. We performed RNA-seq analysis to investigate GA-mediated changes in gene expression profiles in the two radish lines. We identified 313 upregulated, differentially expressed genes (DEGs) and 207 downregulated DEGs in NH-JS2 relative to NH-JS1 in response to GA. Of these, 21 and 8 genes were identified as flowering time and GA-responsive genes, respectively. The results of RNA-seq and quantitative PCR (qPCR) analyses indicated that *RsFT* and *RsSOC1-1* expression levels increased after GA treatment in NH-JS2 plants but not in NH-JS1. These results identified the molecular mechanism underlying differences in the flowering-time genes of NH-JS1 and NH-JS2 after GA treatment under insufficient vernalization conditions.

## 1. Introduction

Gibberellins are tetracyclic diterpene acids that are synthesized in plastids and then translocated into the cytosol in a biologically active form [1]. Bioactive gibberellic acids (GAs) control diverse processes throughout the plant life cycle, encompassing seed germination, stem and leaf growth, trichome development, flowering time [2], and vegetative and reproductive development [3]. GAs are involved in plant growth, development, cell expansion, and division, and respond to specific combinations of internal cues and external stimuli [4,5,6]. Several recent studies reported that GA signaling also is involved in abiotic stress adaptation along with other hormone-signaling pathways [7,8,9]. GAs are a large family of more than 130 structurally related compounds, although only a limited number of GAs display intrinsic biological activity [10]. GAs are synthesized at their sites of action to regulate growth, and GA levels are tightly regulated through a process of feedback regulation to maintain optimal levels for coordinating plant growth and development [11]. Our knowledge of the molecular mechanisms underlying GA signaling in plants was advanced by the following two discoveries: GIBBERELLIN-INSENSITIVE DWARF1 (GID1) encodes a soluble GA receptor, and the DELLA (Asp-Glu-Leu-Leu-Ala) transcriptional regulators negatively control the GA-signaling pathway [12,13]. GID1 binds bioactive GA in a deep binding pocket, and the *N*-terminal extension induces conformational changes that result in covering the GA pocket [12,13]. The GID1-GA complex can bind DELLA to produce the GID1-GA-DELLA protein complex, followed by the ubiquitin-ligase complex Skp, Cullin, F-box containing complex (SCF)-dependent degradation of DELLA protein, ultimately triggering GA-mediated downstream responses [14,15]. Manipulating endogenous GA levels is an established practice in agriculture to modulate plant stature, and the introduction of dwarf alleles into staple crops greatly increases grain yields [16,17]. For this reason, one of the leading aims in the Green Revolution was inducing semi-dwarfism traits, which improved harvest index, enhanced lodging resistance, and increased yield [18,19].

The induction of flowering is the most important event during the transition from the vegetative phase to the reproductive phase during the entire life cycle of higher plants. Flowering induction is precisely regulated by the interplay between endogenous cues and genetic pathways that respond to environmental stimuli such as photoperiod, vernalization, age, and autonomous and gibberellin pathways [20,21]. These signals converge on a small number of floral integrators, including FLOWERING LOCUS T (FT), SUPPRESSOR OF OVEREXPRESSION OF CONSTANS1 (SOC1), and LEAFY (LFY), eventually leading to the activation of floral-meristem identity genes [22]. GAs generally induce bolting and flowering in long-day and biennial plants, although GA is not a universal flowering stimulus [23]. Several studies reported that GAs had complex roles in flowering induction that varied under different circumstances and in different plant species. Under nonpermissive conditions, GAs inhibit flowering in perennial plants but not in long-day and biennial plants [24,25]. It is believed that GAs promote vegetative growth instead of reproductive growth, thereby inhibiting flowering [2]. Although GAs affect flowering in a species-dependent manner, their function in flower development is probably universal. GAs act directly or indirectly to upregulate flowering time gene expression in leaves, and FT protein moves as a mobile signal from the leaf to the shoot apex, where it activates *SOC1* and *LFY* by repressing the DELLA negative transcriptional regulators [26]. In Arabidopsis, GA signaling is crucial for bolting and flowering regardless of the active photoperiod, which was verified in GA-deficient mutants that lacked the GA receptor [23]. GA has a crucial role in flowering under short-day conditions by activating SOC1 when vernalization is not sufficient to induce SOC1 activation, and evidence from GA signaling and biosynthesis mutants shows that SOC1 integrates a GA-dependent flowering pathway [27]. To deepen our understanding of the mechanism that orchestrates flowering in Arabidopsis, we must identify how GA, photoperiod, and vernalization interact. GAs only act under vernalized conditions in plants that require vernalization; nonvernalized plants were unable to bolt or flower in response to GA [28]. In this case, vernalization is a prerequisite for flowering; GA can compensate for photoperiod, but not for vernalization. Vernalization increases the endogenous GA content in Brassicaceae; however, the mechanism mediating GA-induced bolting and flowering has not been completely elucidated.

Radish (*Raphanus sativus* L.) is one of the most important root vegetable crops in the Brassicaceae family and is cultivated worldwide. The tap root contains minerals, vitamins, dietary flavonols, and high glucosinolate content; therefore, the crop’s economic value is primarily determined by root characteristics. Premature bolting and flowering can cause poor root development and serious economic loss in radish crops, particularly in the spring. Agricultural productivity in radish depends on the avoidance of early bolting to produce high-quality leafy vegetables. Studies on the flowering pathways and molecular functions of flowering-related genes have progressed in recent years in model plants, but few studies have focused on radish. The reference radish genome has been sequenced [29,30]. The advent of next-generation sequencing technologies enables genome-wide gene expression profiling and large-scale discovery of flowering-related genes in radish under diverse biological conditions. Recent work investigated a putative model of the bolting and flowering regulatory networks in radish by performing comparative analyses of microRNA (miRNA)-differentially expressed gene (DEG) data from vegetative and reproductive leaves [31]. This study also identified 142 flowering time genes from several developmental tissues using de novo transcriptome analysis. Our group identified 218 radish flowering time genes and a large number of DEGs that responded to vernalization at different bolting times in the late flowering (NH-JS1) and early flowering (NH-JS2) inbred radish lines. We proposed that similar genes were expressed in the vernalization pathways of the two inbred radish lines, and the vernalization pathway was conserved between radish and Arabidopsis [32]. The comparative transcriptome results of the two radish inbred lines with different bolting times suggested a regulatory network of flowering time genes. Vernalization is a key process for bolting and flowering in radish; radish does not bolt or flower without vernalization even when plants are grown for more than 100 days after seed germination [33]. The vernalization threshold requirement is a prerequisite for GA activation of bolting and flowering in radish [34]. Although studies of radish flowering are currently in progress, there are no reports on the mechanism mediating the GA-induced transition from vegetative growth to reproductive growth in radish.

In this study, we performed RNA-seq and qPCR analyses to examine the effect of exogenous GA on global gene expression profiles in two radish inbred lines with different bolting times (NH-JS1 and NH-JS2), and identified the molecular mechanism regulating marginal vernalization.

## 2. Results

### 2.1. GA Effects on Bolting Time Significantly Differed between the Two Radish Inbred Lines

To identify the mechanism of GA-induced bolting and flowering in radish, we observed the effects of exogenous GA on bolting phenotypes in two inbred lines, NH-JS1 (late bolting) and NH-JS2 (early bolting). Seeds were subjected to 0, 0.1, and 10 mM GA during germination, and then the seedlings were vernalized for 0, 10, and 20 days to examine the bolting phenotypes under various vernalization durations. GA did not significantly affect the bolting time in NH-JS1 at 10 days of vernalization. By contrast, GA significantly affected bolting time in the early bolting NH-JS2 line (Figure 1A). In the absence of exogenous GA, neither inbred line bolted even when grown up to 10–50 days after vernalization (DAV) (data not shown). Low GA levels (0.1 mM) can compensate for insufficient vernalization to induce bolting in the NH-JS2 line, but not in the NH-JS1 line. These combined results indicate that NH-JS1 and NH-JS2 differ in their responses to GA and have different bolting characteristics. Statistical analysis of the bolting percentage (the number of bolting plants in each line, (*n* = 20)) is presented in Figure 1B. There were no bolting plants in either line in the absence of vernalization, even when subjected to high GA concentrations. Treatment with 10 days of vernalization and 10 mM GA can induce bolting at 25 DAV, but only in the early-bolting NH-JS2 line. Treatment with 10 days of vernalization and 0.1 or 10 mM GA followed by plant growth for 30–35 DAV resulted in 60% and >80% bolted plants, respectively, in NH-JS2, whereas NH-JS1 plants did not bolt within 30 DAV even with 10 mM GA. Subjecting plants to 20 days of vernalization in the absence of GA resulted in approximately 16% of NH-JS2 plants bolting at 35 DAV, whereas no NH-JS1 plants bolted. Longer vernalization duration enhanced the bolting percentage in both inbred lines in proportion to the GA concentration, although NH-JS1 was much less sensitive to GA than NH-JS2. For example, 0.1 mM GA resulted in 5-fold, 4-fold, and 2.5-fold higher bolting percentage in NH-JS2 than in NH-JS1 at 20, 25, and 30 DAV, respectively (Figure 1B). These results indicate that radish requires at least 20 days of vernalization to bolt, and GA can compensate or induce bolting and flowering under insufficient vernalization. We suggest that marginal vernalization is dominant, and GA modulates the vernalization response during the radish floral transition.

### 2.2. GA Regulation Prioritizes the Floral Transition over Vegetative Growth in Radish

To determine whether GA hyposensitivity in NH-JS1 plants is specific for the bolting trait, we tested the effect of GA on hypocotyl length during seed germination. Figure 2 shows that the phenotypes of GA-treated seedlings were similar in both inbred lines, with growth proportionally enhanced relative to the GA concentrations (Figure 2A). The hypocotyl lengths were measured during a six-day period in the two inbred lines under different GA concentrations. In the presence of GA, hypocotyl lengths in both inbred lines increased gradually with increasing GA concentrations, and were nearly identical in both lines under each condition (Figure 2B). The hypocotyl length was slightly more sensitive to GA in NH-JS1 seedlings than in NH-JS2 at six days after 10 mM GA treatment (NH-JS1, 1.9-fold; NH-JS2, 1.6-fold) (Figure 2B). These combined results in both inbred lines indicate that GA effects on bolting phenotype depend on the duration of vernalization, but not on plant growth.

To identify the precise effect of GA on radish bolting, we observed the GA-induced bolting time phenotype in the two inbred lines after minimal vernalization treatment or no vernalization (control). For this study, 4-day-old seedlings were vernalized for 10 days and acclimatized in a growth room for 14 days. Then, we directly sprayed 10 mM GA on the seedlings and compared the bolting phenotypes in the two lines at 17 days after GA treatment (DAG) (Figure 3A). In the absence of vernalization, none of the plants in the two lines bolted regardless of GA treatment, although both lines exhibited enhanced vegetative growth in response to GA (Figure 3A, 0 days vernalization). A 10-day vernalization treatment combined with exogenous GA treatment produced different bolting phenotypes in the two lines. All NH-JS2 plants bolted under these conditions, and NH-JS2 plants represent typical bolting and flowering characteristics in radish. We suggest that GA efficiently induces the floral transition under the conditions of a required minimum vernalization pretreatment. By contrast, NH-JS1 plants were hyposensitive to exogenous GA in the induction of bolting and flowering, and they did not bolt under the same conditions (Figure 3A, 10 days of vernalization). The effects of GA on plant growth were observable in both inbred lines. Statistical analysis of the bolting percentages (the number of bolting plants in each line (*n* = 20)) in response to GA treatment after 10 days of vernalization are presented in Figure 3B. The bolting percentages in early-bolting NH-JS2 plants differed with and without GA; without GA, approximately 70% of plants bolted at 30 DAG, and then no more plants bolted until 40 DAG, whereas up to 100% of GA-treated plants bolted before 25 DAG. By contrast, approximately 20% of late-bolting NH-JS1 plants treated with GA bolted at 30–40 DAG (Figure 3B). These results suggest that NH-JS1 plants require longer vernalization than NH-JS2, and NH-JS1 plants are less sensitive to GA than NH-JS2. GA can induce the floral transition under insufficient vernalization, but GA cannot independently induce the floral transition without vernalization.

### 2.3. Transcriptome Sequencing Identifies Genes Responding to GA in Radish

We conducted RNA-seq analysis to identify the genes involved in GA-mediated flowering induction in radish. First, we performed qPCR analysis to examine the effective time point of GA application on the expression of major flowering time genes in samples harvested at 6 h, and at 2, 4, and 6 days after GA treatment: 4-day-old seedlings were vernalized for 10 days and transferred to the growth room for 2 weeks to acclimatize, then 10 mM GA solution was sprayed on the leaves. The highest difference of flowering time gene expression level between two lines was observed at 6 days after GA treatment such as *RsMAF2* and *RsSOC1* in accordance with their bolting traits (Appendix A). Therefore, we isolated RNA from shoot tissues, as stated above, collected 6 days after treatment with or without GA to identify global gene expression changes in the two inbred lines in response to GA as shown in Appendix A. A total of eight samples were analyzed from the two inbred lines, with two biological replicates for each condition. We constructed cDNA libraries and sequenced the libraries using an Illumina HiSeq 2000 Sequencing System (Appendix A). A total of 347,399,408 paired-end reads (lengths up to 101 base pair) were produced from eight generated libraries. Raw reads were subjected to quality control, and adapter sequences and low-quality reads were excluded. Approximately 72% clean reads assured the following criteria: Quality score Q > 20 and minimum read length ≥ 25 bp. Ultimately, a total of 251,429,330 clean reads was obtained from the libraries of the two inbred lines selectively treated with or without GA, and the average length of clean reads was 80.06 bp (Appendix A). To verify the similarity between two replicates, normalization counts were used to plot pairs of replicate samples. All samples had good reproducibility between pairs, showing 0.97–0.98 values (Appendix A). To analyze the proportion of unigenes in the eight transcriptome libraries, all clean reads were mapped to the 71,188 unigene reference sets, which was 94.65% (237,984,258) of reads from the two inbred lines selectively treated with GA and mapped to the reference unigenes. Only approximately 5% of all reads were unmapped (Appendix A). These combined results indicate that all of the transcriptome sets retained a high proportion of unigenes and were adaptable for subsequent DEG and expression profiling analyses.

### 2.4. Comparative Analysis of DEGs in the Two Inbred Radish Lines Treated with GA under Marginal Vernalization

To identify the genes involved in regulating GA-induced floral transition and bolting in radish, we performed a genome-wide comparative DEG analysis of the two inbred lines treated with GA. The total set of expressed genes was subjected to DEG analysis using the DESeq package in R. Then, the gene set was analyzed along with individual characteristics of the inbred line transcriptomes treated with GA. DEGs were analyzed using the following criteria: |log2 (fold-change)| ≥ 0.6, false discovery rate (FDR) ≤ 0.01, and read counts ≥ 500. Little flowering time- and GA-related DEGs were confirmed using |log2 (fold-change)| ≥ 1 conditions (data not shown); therefore, the fold-change was reduced to |log2 (fold-change)| ≥ 0.6. A total of 226 DEGs were obtained in each inbred line under GA treatment (GA was sprayed after 10 days of vernalization and 14 days of growth). Of these, 32 upregulated and 71 downregulated DEGs were detected in NH-JS1, and 55 upregulated and 86 downregulated DEGs were detected in NH-JS2 (Appendix A). Only 5 upregulated and 13 downregulated DEGs displayed overlapping regulation between the two inbred lines. The downregulated DEGs were more common than upregulated DEGs in both inbred lines. This result is consistent with the recent DEG analysis of *Rosa chinensis* treated with GA [35].

To identify the active biological pathways in each line in response to GA, we performed functional pathway enrichment analysis by comparing DEGs in NH-JS1_GA vs. NH-JS1 and NH-JS2_GA vs. NH-JS2 using the Kyoto Encyclopedia of Genes and Genomics (KEGG) database. A total of 83 DEGs from NH-JS1 and 103 DEGs from NH-JS2 did not show a significant distribution of specific pathways. We identified three KEGG pathways that contained the highest number of assigned DEGs in each line. Each group of upregulated DEGs were commonly assigned to ‘plant hormone signal transduction’ (NH-JS1 vs. NH-JS2, 7 vs. 4) and ‘metabolic pathway’ (7 vs. 3), whereas downregulated DEGs were commonly assigned to ‘biosynthesis of secondary metabolites’ (6 vs. 8) and ‘metabolic pathways’ (7 vs. 12). The ‘plant-pathogen interaction’ (0 vs. 7) and ‘ribosome’ (0 vs. 6) pathways were matchless pathways whose DEGs were upregulated and downregulated by GA, respectively, in NH-JS2 compared with NH-JS1. The ‘galactose metabolism’ (4 vs. 0) and ‘zeatin biosynthesis’ (2 vs. 0) pathways were unique to NH-JS1 as upregulated and downregulated DEGs, respectively (Appendix A).

### 2.5. Comprehensive Analysis of DEGs Related to GA-Responsive Flowering Pathways in Radish

We analyzed the differences in GA responses between the two inbred lines. A total of 3165 DEGs were identified in the two inbred lines with and without GA treatment. A total of 1324 upregulated and 960 downregulated DEGs were detected in NH-JS2 vs. NH-JS1 with GA treatment, although approximately 80% (1764 DEGs) overlapped with those in NH-JS2 vs. NH-JS1 without GA treatment. Consequently, 313 upregulated and 207 downregulated GA-specific DEGs were detected in NH-JS2 compared with NH-JS1 (Figure 4A). To further evaluate the GA effect on phenotypic variation between the two lines, we conducted statistical enrichment of GA-specific DEGs in the two lines using KEGG pathway analysis. The GA-specific DEGs in the two inbred lines (NH-JS2_GA vs. NH-JS1_GA) were enriched in 13 significant pathways (Appendix A). The highly significant pathways included ‘carotenoid biosynthesis’, ‘biosynthesis of secondary metabolites’, ‘glucosinolate biosynthesis’, ‘arachidonic acid metabolism’, ‘nitrogen metabolism’, and ‘ubiquinone and other terpenoid-quinone biosynthesis’ (*p* < 0.001). In the absence of GA, the NH-JS2 vs. NH-JS1 DEGs primarily mapped to ‘ribosome’ followed by ‘aminoacyl-transfer RNA biosynthesis’, ‘porphyrin and chlorophyll metabolism’, and ‘pantothenate and CoA biosynthesis’ pathways. These combined results indicate that GA effects were indirectly rather than directly related to flowering or GA-related genes in the two inbred radish lines.

To identify flowering time genes that respond to GA and affect bolting and flowering in our radish transcriptome data sets, we examined 218 flowering time genes [32] and 125 GA-related genes using the interactive flowering database FLOWerRing interactive database (FLOR-ID) and published literature on DEGs in flowering-related pathways (Appendix A). We applied the same following criteria: |log2 (fold-change)| ≥ 0.6, false discovery rate (FDR) ≤ 0.01, and read counts ≥ 500. We detected 15 upregulated DEGs and 6 downregulated flowering time DEGs in response to GA in the two inbred lines (Figure 4B, Appendix A). A total of 17 flowering time DEGs in response to GA overlapped with those in the absence of GA in the two lines. *RsELF3* and *RsSOC1 Ft* DEGs were the most strongly upregulated among the GA-responsive DEGs. By contrast, *RsFLC* (Theragen Bio Institute Unigene: TBIU004737) and *RsMAF2*, which are flowering repressors, were downregulated in NH-JS2 with similar expression levels in the absence of GA as in our previous report [32]. Although no significant differences were observed with and without GA treatment, the differential expression of these flowering time DEGs was slightly increased by GA (Appendix A). We detected 6 upregulated and 4 downregulated GA-related DEGs in the two inbred lines that were classified as ‘response to GA’ using the same criteria (Figure 4C, Appendix A). The GA-regulated flowering activator *RsGASA6* [36] was more strongly upregulated in NH-JS2 than in NH-JS1, whereas *RsEXPA1*, *RsPIF4*, and *RsMYB28* or *RsMYB29* were more strongly downregulated in NH-JS2 than in NH-JS1 with GA treatment. The expression of GA-related DEGs *RsGNC*, R*sVI1*, *RsMYB59*, and *RsBETAFRUCT4* was unique to NH-JS2 in response to GA. RNA-seq analysis indicated that only 8 flowering time and GA-related DEGs were differentially expressed in response to GA between the two lines (Figure 4D). We concluded that this result is insufficient to provide insights into the flowering mechanism in radish.

### 2.6. RsFT and RsSOC1-1 Floral Integrators Were Responsive to GA under Marginal Vernalization in Radish Flowering Pathways

To identify the undetected GA-specific flowering time transcripts and corroborate the putative GA-responsive flowering time DEGs, we compared transcript levels of flowering genes in the two inbred lines by performing qPCR analyses. For this purpose, we selected flowering time DEGs that exhibited different expression levels in the presence and absence of GA or were primarily involved in the flowering biological pathway and gibberellin pathways (Figure 5).

First, we identified changes in expression levels of flowering time genes involved in the GA biosynthesis and signaling pathway. The trends of GA-induced increases and decreases in gene expression levels were similar in both lines. However, the transcript fold-changes in *RsGA20ox2*, *RsKAO2*, *RsGID1A*, *RsGAI* (DELLA), and *RsGA2ox2* levels in response to GA differed between the two lines from 1.5-fold up to 5-fold. *KAO2*, *GA20ox2*, and *GA20ox3* are included in the GA biosynthesis pathway in Arabidopsis [37], whereas *GID* and *GAI* [38,39] function in GA-signaling pathway as a GA receptor and GA negative regulator, respectively, belonging to the DELLA family [40,41]. *RsBETAFRUCT4* transcript levels were similar in the presence and absence of GA, unlike a GA-specific DEG identified in RNA-seq analysis (Figure 5A).

To determine the expression levels of flowering time genes involved in the photoperiod pathway, we performed qPCR analysis of *RsELF3*, *RsLHY*, *RsGI*, *RsTEM1*, and *RsCO1 Ft* genes. *RsELF3* did not respond to GA in either line in the qPCR analysis, unlike in the RNA-seq analysis. The morning loop gene [42] *RsLHY* slightly decreased in both inbred lines when treated with GA; eventually, the differential expression levels between the two lines changed in response to GA. Similarly, qPCR analysis of *RsGI* and *RsTEM1* DEG expression levels significantly differed in the two lines in response to GA (1.5-fold up to 2.5-fold), whereas the *RsCO* transcript level did not differ between the two lines in response to GA. *GI* positively regulates *CO* and was increased by GA treatment, and their upregulation patterns were consistent with previous studies [13,43]. TEM1 directly represses expression of the GA_4_ biosynthetic genes *GA3OX1* and *GA3OX2* [44], and acts to immediately repress flowering time gene expression and counteract the activator *CO* gene expression in Arabidopsis [45]. Thus, TEM1 links both photoperiod and gibberellin pathways to control flowering. *RsTEM1* was reduced in response to GA in the same manner that *GA3OX* genes were significantly upregulated in the *tem1* mutant in Arabidopsis. However, the comparative transcript levels in early- and late-flowering inbred lines did not match the bolting traits; *RsTEM1* expression was higher in NH-JS2 than in NH-JS1, and the differential expression levels were slightly larger in response to GA (Figure 5B).

Next, we quantified the expression of vernalization pathway genes. Transcript levels of the flowering repressor *RsFLC1* (TBIU004737) were lower in NH-JS2 in response to GA. Conversely, the transcript level of *RsAGL19*, a repressor of FLC1, was higher in NH-JS2 in the presence of GA. *RsFLC1* (TUBI055229) transcript levels between two lines were not significant to GA as shown in Appendix A. *RsVRN1* expression was strongly increased in late bolting NH-JS1 plants without GA treatment, consistent with our previous RNA-seq results [32]. *RsVRN1* expression in NH-JS1 plants with GA treatment was reduced to a rarely expressed level (about 10-fold), whereas its expression was increased (approximately 2-fold) in NH-JS2 plants with GA treatment. Therefore, *RsVRN1* transcript level was higher in NH-JS2 than NH-JS1 after GA treatment, which may contribute to the observed GA hypersensitivity of NH-JS2 for early bolting and flowering. Expression of the repressor gene *RsMAF2* also responded to GA; it was less sensitive to GA stimulation in NH-JS2, whereas its transcript level was reduced about 4-fold in response to GA treatment in NH-JS1 (Figure 5C).

The expression of flowering time genes involved in the integration of multiple flowering signals was evaluated. The key floral integrator *RsFT* showed significantly different gene expression levels in response to GA in the two inbred lines. There was essentially no difference in the expression level of *RsFT* gene in NH-JS1, whereas transcript levels were more than 3-fold higher in NH-JS2 with GA than without GA. The flowering time integrator *RsSOC1-1* also showed significantly higher expression levels in response to GA treatment in NH-JS2, but were slightly reduced by GA in NH-JS1. The expression level of *RsSOC1-2*, an isoform of *SOC1*, also changed similarly as *RsSOC1-1* in response to GA between the two lines (Figure 5D). *RsSOC1-3* expression levels were not detected in this study (data not shown).

Several GA-specific flowering time genes were quantitatively identified by qPCR analysis. Some flowering time genes were not detected as flowering time- or GA-related DEGs due to RNA-seq limitations, including low abundance (*RsGA2OX2*, *RsGID1A*, *RsTEM1*, and *RsFT*) or differential expression estimation (*RsAGL19*, *RsSOC1-1*, and *RsSOC1-2*) (Appendix A). The qPCR results revealed a transcriptional GA feedback loop that regulates GA primary response genes rather than exogenous GA treatment in radish. Expression of the essential floral genes *RsFLC1*, *RsFT*, and *RsSOC1-1*, which are generally responsible for the floral transition, was consistent with the bolting phenotype induced by GA treatment in radish.

## 3. Discussion

Our previous study confirmed that vernalization-mediated flowering in NH-JS2 (early-bolting phenotype) differs from that of NH-JS1 (late-bolting phenotype) [32]. Here, we investigated bolting processes closely linked to GA action under different vernalization periods. We found that NH-JS2 was more sensitive than NH-JS1 to the effect of vernalization after treatment with different concentrations of GA. To identify the GA-responsive molecular network that regulates the flowering pathway in radish, we performed RNA-seq in the two inbred lines treated with or without exogenous GA. The GA-responsive flowering time DEGs and major flowering time genes, which regulate bolting in Arabidopsis, were biologically confirmed by qPCR analysis. We suggest a gene regulatory network for controlling bolting time in response to GA in radish. Based on this model, we propose that GA promotes the vegetative-to-reproductive transition in radish by upregulating expression of the floral integrators *FT* and *SOC1* under insufficient vernalization conditions. The GA effect on flowering pathways is moderately conserved between radish and Arabidopsis.

### 3.1. The Two Inbred Radish Lines Display Different Bolting Times in Response to GA Treatment

GA is a positive plant growth regulator that speeds up bolting and flowering in many species [10,23,28,46] including Arabidopsis [47]. However, the effects of GA on bolting time and the molecular mechanism of GA-mediating flowering had not been reported in radish. To determine how bolting responds to GA under different vernalization periods in radish, we treated seeds of two inbred lines with GA and found that NH-JS2 displayed more bolting than NH-JS1 under insufficient vernalization conditions. In NH-JS1, bolting did not occur when vernalization was relatively short (10 days) even with GA treatment; however, bolting did occur when vernalization was increased to 20 days with the same GA concentration (Figure 1). By contrast, the vegetative growth responses of both inbred lines to GA were essentially the same (Figure 2). GA significantly promoted bolting in NH-JS2 under insufficient vernalization conditions (10 days), but NH-JS1 displayed less bolting under the same conditions (Figure 3). This result indicates that GA has a stronger effect on the early bolting NH-JS2 line, but the late bolting NH-JS1 line reacts more sensitively to vernalization period than exogenous GA treatment. These combined results demonstrate that vernalization is an indispensable factor, whereas GA is likely to have a causal role for bolting and flowering in radish. Similarly, the grass *Lolium perenne* requires both vernalization and long-day conditions for inflorescence initiation, whereas GA promotes bolting in vernalized plants and does not affect nonvernalized plants [28]. Vernalization is essential for bolting and flowering in cauliflower (*Brassica oleracea* var. botrytis), and flowering did not occur in nonvernalized plants even with sufficient GA application [48].

### 3.2. GA-Responsive Flowering Time DEGs in Radish

In recent GA-responsive transcriptome studies, the most prominent gene expression changes in different tissues of *Populus tomentosa* and *Jatropha curcas* plants occurred 6 h after GA treatment [49,50]. Before performing RNA-seq analysis, we collected four samples at 6 h, and at 2, 4, and 6 days after GA treatment to check the expression levels of major flowering time genes using qPCR analysis. The differences in expression of key flowering gene such as *RsSOC1* was prominent on the sixth day after GA treatment between the two lines (Appendix A), so RNA-seq analysis was performed on the sixth day after GA treatment (Appendix A). Although the bolting phenotypes induced by GA are distinct, GA-responsive flowering time DEGs were rare even though |log_2_ (fold-change)| ≥ 0.6, FDR ≤ 0.01, and read count ≥ 500 were used as criteria (Figure 4). Flowering time genes could be assumed to be expressed at very low levels in response to GA; this is consistent with results in other crops as transcriptomes were analyzed without read counts involving flowering time DEGs [51,52,53]. We obtained 21 GA-responsive flowering time DEGs in NH-JS2 vs. NH-JS1. The numbers of upregulated vs. downregulated flowering time DEGs differed by approximately 3-fold between the two lines (upregulated vs. downregulated flowering time DEGs, 15:6) (Appendix A). Although RNA-seq is a powerful tool, there are limits for accessing low abundance transcripts, managing biological variation, and estimating differential expression. Thus, some transcripts may not be captured in the final set of reads. To overcome these limits of detection for certain genes, we conducted qPCR analysis of GA-responsive flowering time genes. Most of the detected GA-responsive flowering time DEGs were similar in the qPCR results, but fold-differences were observed due to normalization (Figure 5). In particular, qPCR analysis of the GA pathway flowering time DEGs (*RsGA20ox2*, *RsGA2ox2*, *RsKAO2*, and *RsGAI*) revealed greater differences in expression levels in response to GA in the two inbred lines than those detected in RNA-seq analysis. The qPCR results for the vernalization pathway gene *RsVRN1* showed remarkable differences in response to GA in NH-JS1, but little differences in NH-JS2. Both qPCR and RNA-seq identified significant differences in expression levels of the key floral genes *RsFT* and *RsSOC1-1* in the two lines according to the bolting traits under GA treatment (Figure 5 and Appendix A). Our results indicate that qPCR analysis helped to provide deeper insights into the GA-responsive characteristics of radish. Further studies are needed to identify the specific functions of these genes and their molecular networks in the transition from vegetative to reproductive development in radish.

### 3.3. A Gene Regulatory Network Model for GA-Responsive Flowering in Radish

It is crucial to identify the molecular mechanism mediating GA responses to determine how GA application integrates the activation of flowering pathways, although it is known that GA accelerates flowering by degrading DELLA repressors. We utilized our qPCR results to develop a model for the three main flowering pathways, gibberellin, vernalization, and photoperiod, under GA treatment in radish (Figure 6). These three pathways converge on flowering integrators. By analyzing GA pathway genes, we identified that activators of GA signaling and biosynthesis were downregulated in response to GA application, whereas repressors were upregulated under low GA levels, indicating a transcriptional GA feedback mechanism functions in radish and other plant species [54]. Our results indicate that the expression of both GA biosynthesis and signaling genes was sensitive to GA, and GA maintains homeostasis through a strict mechanism in radish. In the radish photoperiod pathway, GA stimulates bolting through *RsLHY*, *RsGI*, and *RsTEM1* as activators and a repressor, respectively. LATE ELONGATED HYPOCOTYL (LHY) is a core component of the circadian oscillator [55]. GIGANTIA (GI) has roles in induction of photoperiodic flowering through FLAVIN-BINDING KELCH REPEAT, F-BOX1 (FKF1) protein interaction [56] and functions as a chaperon in the maturation of photoreceptor ZEITLUPE (ZTL) [57], whereas the TEMPRANILLO1 (TEM1) transcription factor negatively regulates GA biosynthesis and directly represses FLOWERING LOCUS T (FT) transcription [45]. Although the expression level of these genes whose expressions are regulated by circadian rhythm was determined at the one time point as 10 a.m., our result shows that GA stimulates the photoperiod pathway activator *RsGI* and represses the *RsTEM1* floral repressor in radish. In the radish vernalization pathway, *RsVRN1* was the most GA-responsive and unique gene that displayed expression pattern changes under GA treatment, consistent with the bolting characters in both lines. The transcript levels declined in response to GA in the late-bolting NH-JS1 line (approximately 14-fold change), whereas the early-bolting line NH-JS2 showed inversely increased expression in response to GA (approximately 2-fold change). Vernalization1 (VRN1) is central to the vernalization response and important to maintain repression of the flowering repressor VRN2, ultimately promoting the upregulation of flowering time expression in Arabidopsis [58]. This result suggests that changes in *RsVRN1* expression in response to GA could largely induce the transition from vegetative to reproductive growth in radish under marginal vernalization conditions. Each flowering pathway was distinct or interrelated, and eventually converged to the floral integrators FT, SOC1, and LFY. Ultimately, upregulation of *RsFT* and *RsSOC1-1* in response to GA is sufficient to induce bolting. *RsLFY* was rarely expressed in both RNA-seq and qPCR analyses in this study. Our RNA-seq results confirmed that most of the flowering gene expression levels were very low, which could be due to two reasons: (1) We performed RNA-seq analysis of young radish shoots, and (2) GA-mediated flowering induction is likely to be effective even at very low levels of gene expression. This study identified transcripts that have differentially low expression levels in response to GA in early-bolting and late-bolting inbred radish lines. These DEGs may reveal crucial information about GA-mediated flowering in radish (Figure 6).

## 4. Materials and Methods

### 4.1. Plant Materials, Exogenous GA Treatment, and Bolting Trait Analysis

The NH-JS1 (late-bolting) and NH-JS2 (early-bolting) radish inbred lines were developed by NongHyup Seed (Geonggi-do, Anseong, Korea) [32]. Seeds of each line were sterilized by soaking in 70% ethanol and 10% chlorine bleach for 30 min to disinfect the seed coat, and then washed with sterile distilled water at least three times. To investigate the effect of exogenous GA on seedlings, seeds of each line were sown on filter paper treated with 5 mL of 0, 0.1, or 10 mM GA (Duchefa Biochemie, Haarlem, The Netherlands) at 25 °C for 1 day. Seedlings with a length of 1–2 mm were grown in the growth room under long-day conditions (25 °C, 16 h light/8 h dark photoperiod at 100 µmol m^−2^s^−1^) for 14 days. Thereafter, the seedlings were vernalized in a cold room (5 ± 1 °C, 12 h light/12 h dark at 100 µmol m^−2^s^−1^) for 0, 10, and 20 days, and then transferred to sterilized soil and grown under normal growth conditions. Twenty seedlings were used for each bolting test. To test the effect of exogenous GA on young plants, 4-day-old seedlings were vernalized (as described above) for 10 days and then transferred to the growth room for 2 weeks to acclimatize. After acclimation, 10 mM GA solution was sprayed directly on the leaves, and plants were grown in the growth room for up to 50 days after spraying. Plants were examined for bolting at 0, 15, 20, 25, 30, 35, and 40 days after vernalization (DAV); more than 20 plants were used for each test. The percentage of bolted plants was calculated by counting the number of plants with floral axis lengths ≥1 cm relative to the plants without floral axes. The bolting percentages for each inbred line after GA treatment of seeds were calculated at 25, 30, and 35 days after each vernalization treatment of 0, 10, and 20 days. The bolting percentages for each inbred line after GA treatment of young plant leaves were calculated at 0, 15, 20, 25, 30, 35, and 40 days after the 10-day vernalization treatment.

### 4.2. Evaluating the GA Effect on Hypocotyl Elongation

Twenty seeds of each inbred line were disinfected, sown in sterilized soil, placed in the dark in a cold room at 5 ± 1 °C for 3 days to enhance germination, and then transferred to a growth room. Radish sprouts with hypocotyls longer than 0.5 cm in length were transferred to another pot containing sterilized soil in the growth room. Then, 1 mL of 0, 1, or 10 mM GA solution was directly poured over each sprout. Hypocotyl lengths were measured every day for 6 days after the GA treatment. More than 20 seedlings of each inbred line were used for hypocotyl measurement at each GA concentration.

### 4.3. Preparation and Sequencing of the RNA-Seq Library

A total of eight shoot tissue samples (two inbred lines × two treatments × two biological replicates) were collected at the same point in the light/dark cycle as 10 a.m. time point, immediately frozen in liquid nitrogen, and stored at –80 °C until further processing. Shoot tissues from three different plants were pooled to obtain sufficient RNA for each extraction. For RNA-seq library construction, total RNA was isolated from shoot tissue and cDNA was synthesized as described by Jung et al. [32]. The RNA-seq libraries were created using the Illumina TruSeq RNA Library Prep Kit (Illumina, San Diego, CA, USA) according to the manufacturer’s protocol. The RNA-seq library was PCR-amplified and sequenced on an Illumina HiSeq 2000 platform. A 101-bp paired-end sequencing protocol was used, and two biological repeats were performed for each sample. All raw, read data generated in this study were deposited in the Gene Expression Omnibus (GEO) functional genomics data repository of National Center for Biotechnology Information under accession number GSE125875: lists the GEO DataSeries.

### 4.4. Reference-Guided Assembly and Mapping of the Radish Transcriptome

Raw sequencing data were filtered to standard Illumina pipeline RNA-seq parameters. Paired-end reads were quality trimmed, and adapter contamination, low-quality parts, and N-base reads were removed. Reads that fell below a Phred quality score (*Q* ≤ 20) and reads shorter than 25 base pairs (bp) were discarded. These steps were performed using the DynamicTrim and LengthSort programs of the SolexaQA (v.1.13) package [59]. Next, the purified paired-end reads were pooled and mapped against available radish reference gene datasets. A reference-based assembly was performed by utilizing 46,512 genes from the coding sequence regions of the radish reference genome to obtain an assembly dataset [30]. Bowtie2 (v.2.1.0) was used to map the purified datasets [60]. The program allows only remarkable mapping, which has a maximum of two mismatches. Otherwise, the default options were used. The expression levels of each sample were determined using in-house scripts. Read counts of each gene were normalized with respect to library size and counted to the nearest whole number.

### 4.5. Functional Annotation Analysis

RNA-seq transcripts were annotated by comparison with gene sequences in the Phytozome database using Basic Local Alignment Search Tool: protein BLAST (BLASTp) with expect values (E-values) that were at least higher than 1E^−10^ (BLAST v.2.2.28+) [61]. The Gene Ontology (GO) database was utilized for GO analysis, and the transcripts were annotated through the GO database using BLASTP (E-value ≤ 1E^−06^). GO term annotation was conducted using GO classification results from the Map2Slim.pl script. Protein sequences were annotated with the highest sequence similarities and cutoffs, and retrieved for analysis. Data for Annotation, Visulization and Integrated Discovery (DAVID) was used for functional enrichment analysis [62]. The transcript lists also were annotated using the The Arabidopsis Information Resource database and clarified according to default criteria (counts ≥ 2 and the Expression Analysis Systematic Explorer score ≤ 0.1). The Kyoto Encyclopedia of Gene and Genome (KEGG) pathways database was used to analyze the sequences using the single-directional best-hit method and the KEGG Automatic Annotation Server [63]. KEGG enrichment analysis was performed as described previously [55].

### 4.6. Analysis of DEGs

Gene expression data were generated from eight samples of the two inbred lines. To identify DEGs, the plant samples were treated with GA for 6 days with 10 days of vernalization. Raw counts were normalized and analyzed using the differntial gene expression analysis based on the negative binomial distribution (DESeq) library in R (v3.2) [64]. Then, DEGs were considered to display fold-change more than |log_2_ (fold-change)| ≥ 0.6 and were filtered by requiring the *p*-value adjustment to be ≤0.01. The control was NH-JS1 (without GA).

### 4.7. Identification of Flowering Time- and GA-Related Genes in Radish

To identify flowering time- and GA pathway-related genes in our transcriptomes, two sets of 218 flowering time and 125 GA pathway-related genes were selected as reference sets based on published literature and studies in *Arabidopsis thaliana* [65,66,67,68] as described previously [32]. Published sequences were obtained from the TAIR database based on Arabidopsis accession numbers for flowering time- and GA-related genes. BLASTn was used to query the 218 flowering time- and 125 GA-related genes against the assembled 71,188 genes of radish. Top hits were filtered based on the highest percentage of hit coverage and sequence similarity. Cutoffs were E-values ≤ 1E^−25^ and identity ≥ 65%. The flowering time- and GA-related genes in Arabidopsis were compared with flowering time- and GA-related gene sequences in radish using BLASTn (E-values ≤ 1E^−25^, identity ≥ 70%).

### 4.8. Quantitative PCR (qPCR) Analysis

To validate the DEGs identified in RNA-seq analysis, we conducted qPCR analysis. Total RNA was isolated from radish with or without GA treatment using RNAiso Plus (TaKaRa, Tokyo, Japan). Total RNA with RNase-free DNase I (Fermentas, Burlington, Canada) was used for cDNA synthesis (RevertAid First-Strand cDNA Synthesis Kit, Fermentas). Then, qPCR was performed in a CFX Connect^TM^ Real-Time PCR System (Bio-Rad, Hercules, CA, USA) using SYBR Premix Ex-Taq (TaKaRa, Tokyo, Japan) according to the manufacturer’s instructions. Relative expression levels were obtained after normalization with radish actin gene (*RsACT*) expression levels. All qPCR experiments were performed using the *flowering time* gene-specific primer set (Appendix A) with two biological replicates, each with three technical repeats, under the same conditions.

## Figures and Tables

**Figure 1 plants-09-00594-f001:**
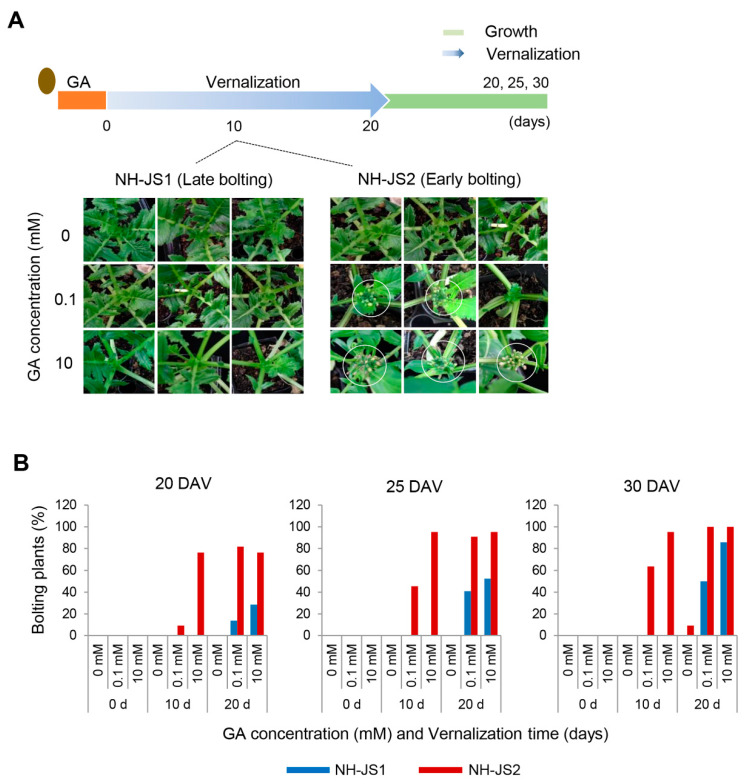
Phenotypes of NH-JS1 and NH-JS2 inbred radish lines under vernalization after gibberellin treatment. (**A**) Bolting phenotypes of NH-JS1 and NH-JS2 under vernalization for 10 days following gibberellin (GA) treatment (0, 0.1, or 10 mM) during seed germination. Seeds were germinated on filter paper in the presence or absence of GA and grown at 25 °C in a growth room. Germinated sprouts were vernalized by transferring into a cold room (5 ± 1 °C, 12 h light/12 h dark) for 0, 10, or 20 days. (**B**) Percentage of bolting radish plants after vernalization (*n* = 20 plants). DAV, days after vernalization.

**Figure 2 plants-09-00594-f002:**
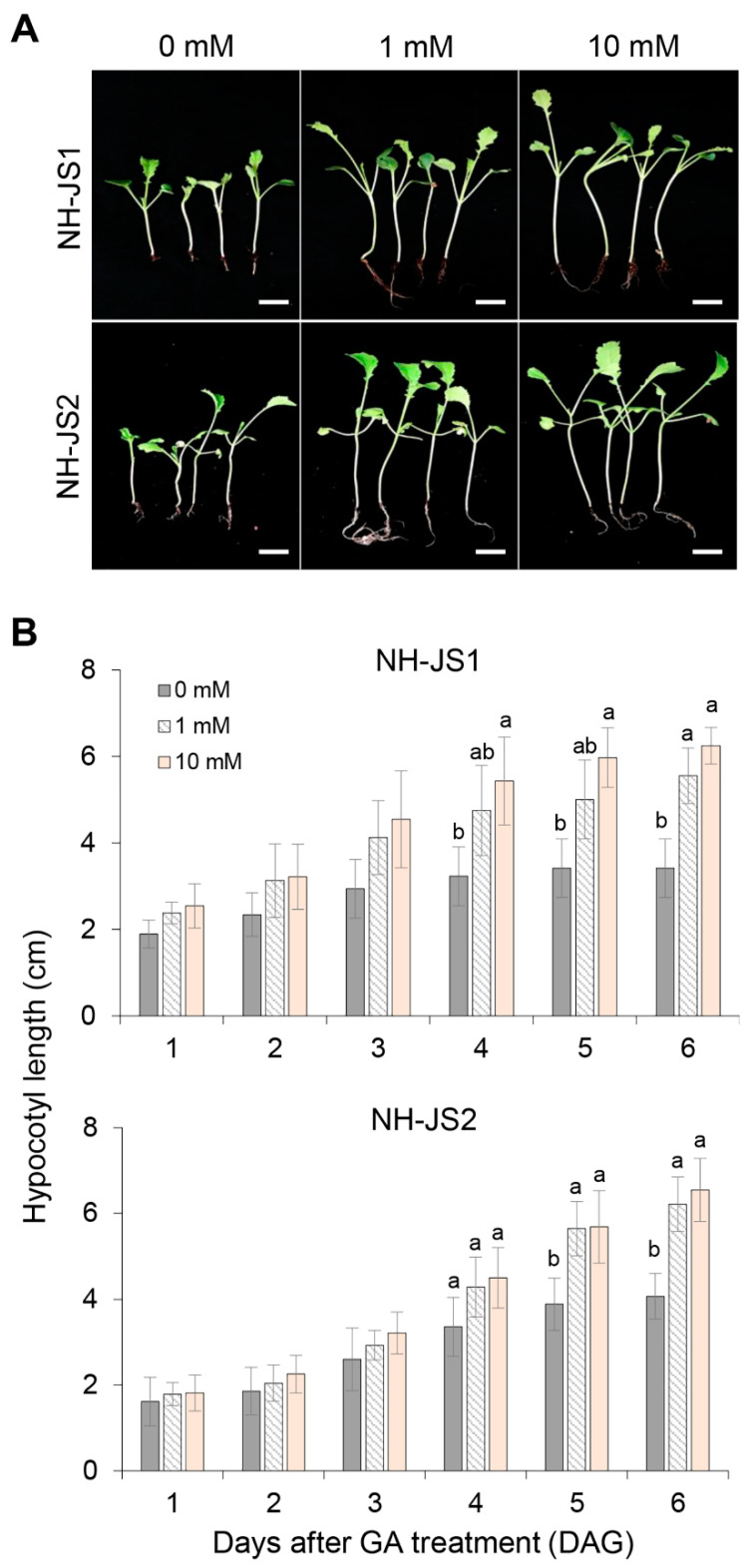
Hypocotyl phenotypes of the two inbred radish lines after GA treatment. (**A**) Hypocotyl phenotypes of the two radish lines after GA (0, 1, or 10 mM) treatment. Seeds were grown in a dark cold room at 5 ± 1 ℃ for 3 days and then transferred into a 25 ℃ growth room. Radish sprouts with hypocotyls longer than 0.5 cm were selected, and GA solution was poured directly over the seedlings (*n* = 6). Hypocotyl phenotypes were observed at 6 days after GA treatment. Scale bars = 0.5 cm. (**B**) Average hypocotyl lengths of NH-JS1 and NH-JS2. Bars represent the average hypocotyl lengths at different GA concentrations. Data are presented as mean ± SD (*n* = 6, except for 3–4 maximum and minimum seedlings). Asterisks mark significant differences between GA treatment compared with the absence of exogenous GA (one-way ANOVA followed by Bonferroni post hoc test).

**Figure 3 plants-09-00594-f003:**
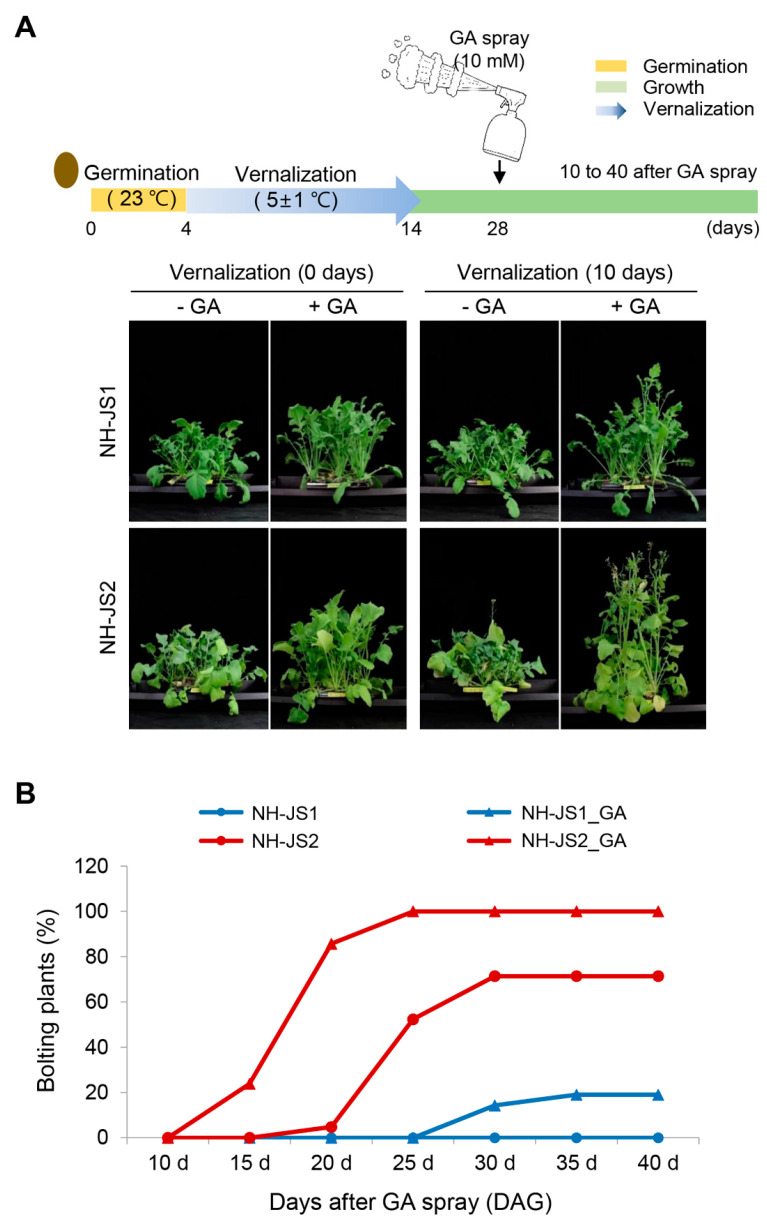
Phenotypes of the two inbred lines sprayed with gibberellin after vernalization. (**A**) Bolting phenotypes of the two inbred radish lines treated with GA (0 or 10 mM) after 10 days of vernalization. Four-day-old seedlings were vernalized for 10 days and then transferred into a 25 °C growth room for 2 weeks. Then, 10 mM GA solution was sprayed directly on the leaves (*n* = 20 plants). Plants were photographed at 17 days after GA spray. (**B**) Percentage of bolting radish plants after GA spray. The number of bolting plants was counted the next day after GA treatment. DAG, days after GA treatment.

**Figure 4 plants-09-00594-f004:**
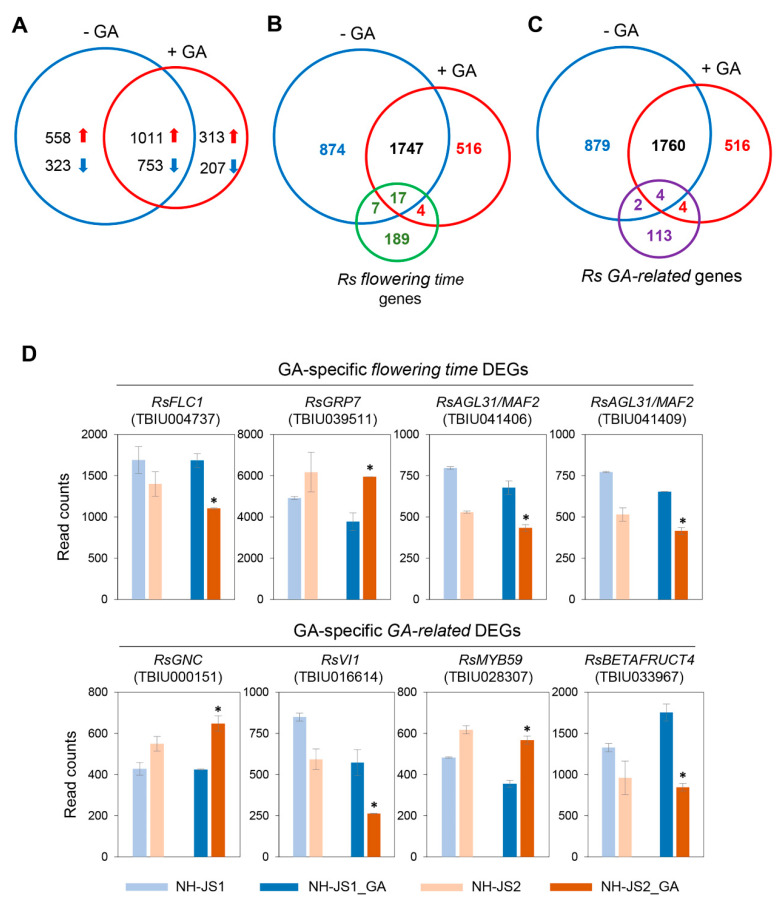
Comparative analysis of differentially expressed genes (DEGs) with and without gibberellin treatment in the two inbred radish lines (|log_2_FC ≥ 0.6|, false discovery rate (FDR) ≤ 0.01, read count ≥ 500). (**A**) Number of DEGs in the two lines with and without GA treatment. Red arrows, number of upregulated DEGs; blue arrows, number of downregulated DEGs. (**B**) The number of flowering time DEGs between two lines under GA treatment. Red, the number of flowering time DEGs after GA treatment. (**C**) The number of GA-related DEGs between two lines under GA treatment. Red, the number of GA-related DEGs after GA treatment. (**D**) RNA-sequencing results of flowering time and GA-specific DEGs. Asterisk means selected DEGs response to GA.

**Figure 5 plants-09-00594-f005:**
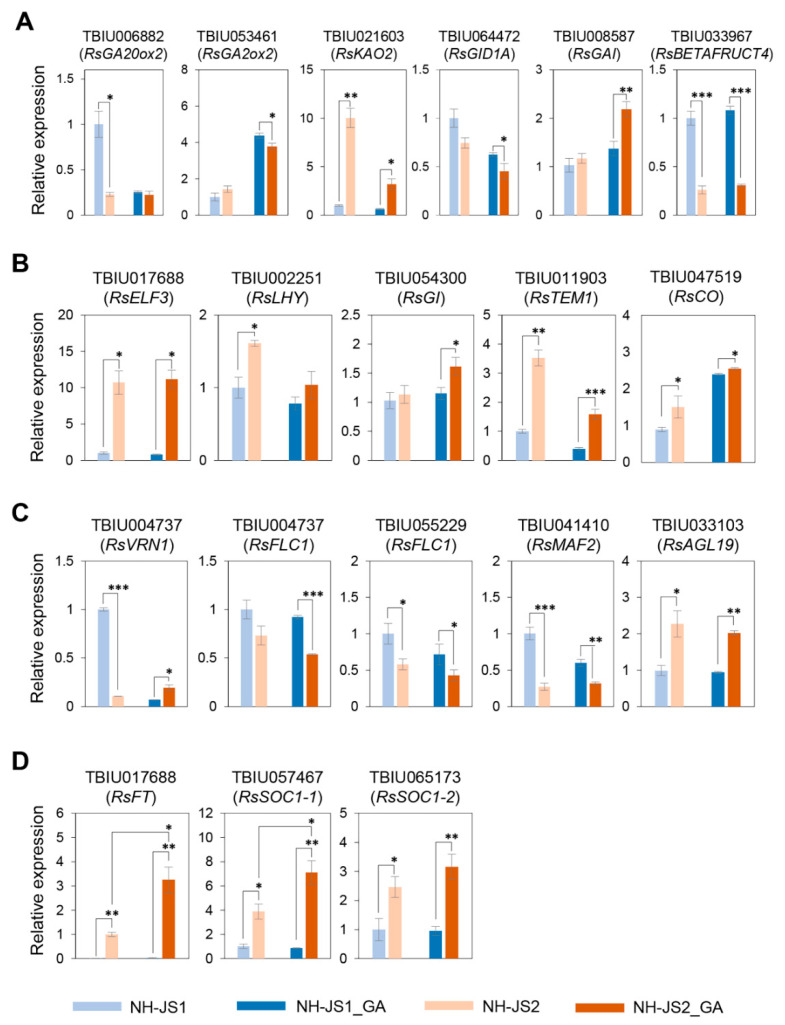
Quantitative PCR analysis of flowering time DEGs in the two inbred radish lines in response to gibberellin. Total RNA was isolated from shoots of NH-JS1 and NH-JS2 inbred lines 6 days after treatment with or without GA (0 or 10 mM) spray application. The complementary DNA was synthesized from total RNA. Upregulated and downregulated genes are grouped according to biological pathways determined from gene ontology analysis. Flowering time genes involved in GA pathway (**A**), photoperiod pathway (**B**), vernalization pathway (**C**) and flowering integrators (**D**). The qPCR values were normalized relative to *RsACT1* (actin) expression level. Error bars represent ± standard error of biological triplicates. Asterisks indicate statistically significant differences (NH-JS2 vs. NH-JS1 and NH-JS2_GA vs. NH-JS1_GA; Student’s *t* test; ** p* < 0.05, *** p* < 0.01, and **** p* < 0.005; two-tailed Student’s *t*-test).

**Figure 6 plants-09-00594-f006:**
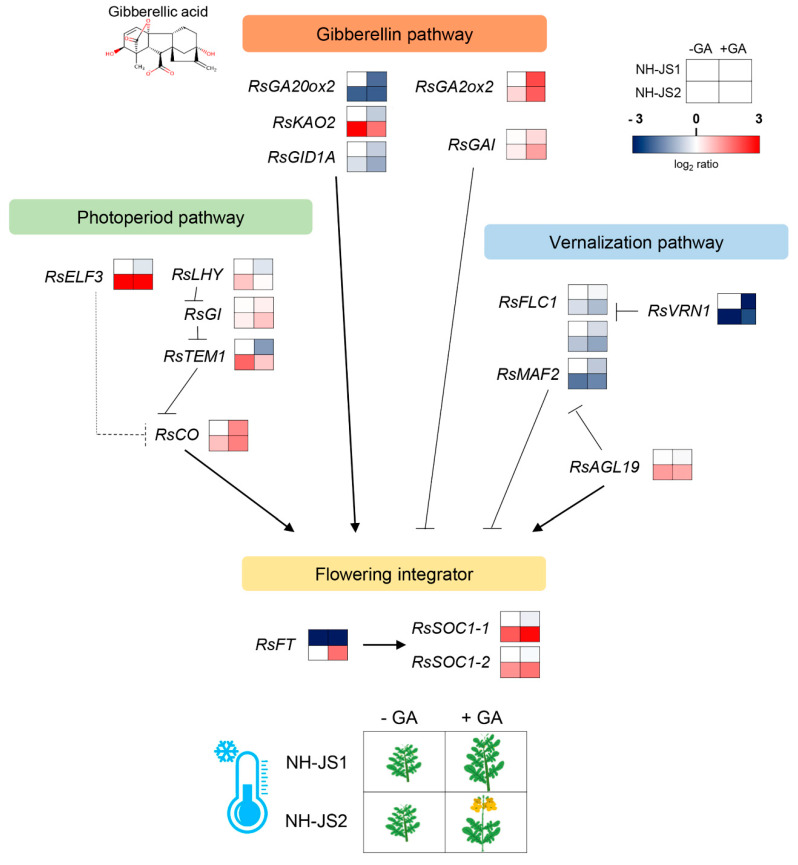
Gene regulatory network controlling gibberellin-accelerated flowering in radish. The illustration maps the regulatory network of flowering time genes in NH-JS1 (late-bolting) and NH-JS2 (early-bolting) plants treated with GA. The network is based on data confirmed by qPCR. Gene expression levels were normalized relative to the expression levels in non-GA-treated NH-JS1 plants (for *RsFT*, non-GA-treated NH-JS2 plant data analysis included log_2_ ratio.). Red indicates higher expression levels and blue indicates lower expression levels relative to non-GA-treated NH-JS1 (*RsFT* expression is relative to non-GA-treated NH-JS2). Arrows indicate transcriptional activation; bars indicate transcriptional repression.

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
