# Peer review of "Gibberellin Promotes Bolting and Flowering via the Floral Integrators *RsFT* and *RsSOC1-1* under Marginal Vernalization in Radish"

_plants, 2020, doi:10.3390/plants9050594_

Round 1
Reviewer 1 Report
The authors describe a series of experiments providing the role of gibberellin in the control of flowering via FT and SOC1 in radish. This manuscript gives some novel insights in the transition from vegetative to reproductive phase in important root vegetable crop. I followed the insistence of the authors. However, several points should be revised before publication.
- Although the author uses “Ft” as a flowering time, the term is not common. Further, the term reads misunderstanding with FLOLING LOCUS T (FT).
- In keywords, RsFT and RsSOC1 seemed to be BrFT and BrSOC1, respectively.
- I understand that DEGs experiment is important to identify the candidate genes regulating flowering in radish. The result section related to DEGs should focus on the flowering and GA related genes differently induced or repressed by gibberellin in two inbred lines. Therefore, I do not understand the importance of metabolism related genes listed in Fig. 4. The author should focus on the genes corelating with flowering time listed in Fig. 6 and reconsider the contents of the manuscript.
- The genes encoding central components of circadian clock show diurnal oscillation. LHY and GI expressions peak at dawn and dusk, respectively. Further, CO expression at dusk is important to induce FT expression in Arabidopsis. Meanwhile the transcription level is restricted in only one point (Fig. 6). The author should state the time point and carefully discuss the transcription level of these genes.
Author Response
Dear Editor;
Enclosed please find our revised manuscript entitled “Gibberellin Promotes Bolting and Flowering via the Floral Integrators RsFT and RsSOC1-1 in Radish” (Manuscript ID: 773574). We have revised the manuscript in accordance with your and the reviewer’s comments. We have changed main figures Figure 2, 3, 4, 5 and Supplementary Figure S4 and S5. The Figure 2 and 3 exchanged and the Fig. 4A combined with Fig. 5, so the number of main figure is 6. The Supplementary Fig. S4 (new one) is added, so the number of supplementary figure is 5.
Referee’s comments (Reviewer 1)
Thank you very much for your precious reviewing. I have responded to each your comments below.
- Although the author uses “Ft” as a flowering time, the term is not common. Further, the term reads misunderstanding with FLOLING LOCUS T (FT).
® I have changed “Ft” to “flowering time” throughout this manuscript for avoiding confused with floral gene FT in accordance with your comment.
- In keywords, RsFT and RsSOC1 seemed to be BrFT and BrSOC1, respectively.
® I have corrected the BrFT and BrSOC1 keywords to RsFT and RsSOC1
- I understand that DEGs experiment is important to identify the candidate genes regulating flowering in radish. The result section related to DEGs should focus on the flowering and GA related genes differently induced or repressed by gibberellin in two inbred lines. Therefore, I do not understand the importance of metabolism related genes listed in Fig. 4. The author should focus on the genes correlating with flowering time listed in Fig. 6 and reconsider the contents of the manuscript.
® We have changed the Fig. 4B to Supplementary Fig. S4 to guard against meaningless result and
Fig. 4A combined with Fig. 5 results in a figure (Fig. 4) as your suggestion.
- The genes encoding central components of circadian clock show diurnal oscillation. LHY and GI expressions peak at dawn and dusk, respectively. Further, CO expression at dusk is important to induce FT expression in Arabidopsis. Meanwhile the transcription level is restricted in only one point (Fig. 6). The author should state the time point and carefully discuss the transcription level of these genes.
® I have described the time point of flowering gene expression as 10 am in the Method parts and mentioned genes involved in circadian rhythms in the Discussion (line: 504-505; 566).
I hope that the changes we’ have made in the manuscript will meet with your approval and will allay the concerns of the Reviewers.
I look forward to hearing from you at your earliest convenience. Please do not hesitate to contact m again if you have any other queries.
Sincerely yours,
Hye Sun Cho
Plant Systems Engineering Research Center
Korea Research Institute of Bioscience and Biotechnology
Daejeon, Korea
Phone: 82-42-860-4469
Fax: 82-42-860-4608
E-mail: hscho@kribb.re.kr

Reviewer 2 Report
Gibberellin Promotes Bolting and Flowering via the Floral Integrators RsFT and RsSOC1-1 in Radish
In the above manuscript Jung et al attempted to persuade that Gibberellin (GA) promotes bolting/ flowering in radish through RsFT and RsSOC1. Research in the model plant Arabidopsis contributed immensely to our understanding of the physiological and genetic pathways involved in regulating flowering time and GA mediated pathways. It is also well known that the findings in Arabidopsis is not always conserved in other plant species especially in crop plants. Therefore, investigations such as the one undertook by the authors in radish, are important to understand the level of conservation of these genetic pathways in other species. It is also important for species specific crop breeding.
Authors started by showing that exogenous GA application differentially promote bolting in vernalised plants using a late (NH-JS1) and early (NH-JS2) flowering radish lines (Figure 2). Then authors went on to perform a transcriptome analysis using RNAseq to find differentially expressed genes between these two lines and GA treated and untreated plants. Finally, they used qPCR to validate RNAseq data and propose a molecular model for the GA promotion of bolting in radish.
I do have the following major and minor concerns.
Major Concerns
- The major conclusions of the manuscript is mainly based on the RNAseq data and the data has many major flaws as follows.
- Q >=30 is the generally accepted norm for RNAseq read quality cutoff. The authors used Q >=20 (I believe it is a typo that authors mention that they used less than (Q <=20) sign?) and read length >=25 bp. Therefore, quality of the reads seems generally low.
- Authors have used a single arbitrary time point without compelling evidence to support their choice. Authors chose 6 Days after GA treatment and the justification was (Line 210) “The highest Ft gene expression level was observed at 6 days after GA treatment (Supplementary Figure S1)”. Although, it’s not clear which is that “Ft gene” authors referred to, it is clear (Figure S1) that the transcript levels of many genes tested were increasing with time. How did the authors decide not to monitor the transcript levels after 6 days to decide whether there is another better time point further down
- Moreover, the transcript level changes of the major genes are not convincingly GA-dependent and suggest vernalization of 10 days alone can cause such changes (Figures 5 and 6: NH-JS2 vs. NH-JS2_GA). The enriched pathways are all secondary metabolite pathways and not even indirectly related to flowering (Figure 4). If RsFT and RsSOC1 are direct targets of GA then, authors should have seen them as top differentially expressed genes in the GA treated plants. However, RsFT wasn’t significant and RsSOC1 wasn’t any different from untreated plants (Table S4). Authors have to perform a qPCR analysis to find whether FT and SOC levels are differentially altered. Nevertheless, authors qPCR data has consistency issues: Figure S1, RsFLC1 levels go up in both GA treated plant lines compared to untreated plants at 6d and not much different between the two lines. In Figure 6 however, RsFLC levels were slightly down in NH-JS2_GA.
- Furthermore, FT and SOC1 are downstream floral integrators and the pathway which promoted flowering won’t be reflected by their transcript levels. Cause-effect relationships cannot be established by transcript levels.
Same authors have reported in reference 32 that 35 days of vernalization promotes bolting to 100 % in both lines 30 DAV without the need for GA treatment. This implies that GA mediated promotion of bolting may not be direct, and GA promotes flowering by promoting growth. This is further evident by the fact that even treating the seeds with 10 mM GA and then vernalize for 10 days (also see comment 2 below) could promote flowering almost up to 80% in NH-JS2 (Figure 1B).
If the authors would want to make the claim that GA directly promotes flowering via RsFT and RsSOC1 as they did in the manuscript, they need an RNAseq time course to show that at least FT and SOC1 levels change meaningfully with time in GA dependent manner. Otherwise, authors need to modify the manuscript to admit the above reality.
- Figure1: Seeds were treated with GA. How persistent (non-degradable) is GA to be responsible for promoting bolting many days after germination? The observed differences among treatments could be simply due to promotion of germination and early growth, by GA and therefore the treated plants responded differentially to 10 d and 20 d vernalization treatments. Figure 3 suggests it is the case. Figure 3 should be Figure 2 to explain that fact. Differences observed in the two lines (NH-JS1 and NH-JS2) is simply inherent response to vernalization treatments. Reference 32 showed that the two lines respond differentially to vernalization treatment.
- Line 174: Authors claim that “but GA cannot independently induce the floral transition without vernalization”. However, authors did not include appropriate controls in Figure 2 to make that claim. Authors need NH-JS1 and NH-JS2 plants without vernalization and with or without GA treatment. Controls in Figure1 are not relevant for Figure2 (see comment 2 above).
- Lines 185-186: Authors state that “Therefore, we conclude that GA treatment differentially affects Ft-related gene expression in the two inbred radish lines “. At this point authors haven't shown any data yet. Authors may use “hypothesize” instead of can't “conclude”.
- Lines 319-320: Based on the RNAseq data authors claim that “This suggests that regulators of GA biosynthesis and signaling are involved in bolting and flowering pathways as activators or repressors in radish.”. Authors need genetic evidence to make such strong claims.
- RNAseq data set is not found in the repository (GEO functional genomics data repository of NCBI under accession number SRP176394) and needs to be submitted before publication.
Minor Concerns
- Authors use an unintroduced abbreviation “Ft” throughout the manuscript: Is this Flowering Locus T (FT) or flowering time genes? Please do not use confusing terminology not used generally in the field. In some places, authors use Ft gene (Lines 75, 210) which confuses even more as to whether this is one gene authors talk about.
- Figure 1B: How is it possible to have more than 100%? Why is there a 120 in the y-axis?
- For each Figure / experiment authors need to mention that how many plants were measured and the statistics. e.g. missing in Figure 2B.
- Plant samples for RNAseq (Lines 208-210): Missing a lot of important information. Information in Fig. S2 should be highlighted here if the samples were treated in a similar way.
- Supplementary Table S2 is missing.
- RNAseq DEG cutoff (Lines 235-236): “DEGs were analyzed using the following criteria: log2 (fold-change) ≥ 0.6, false discovery rate (FDR) ≤ 0.01, and read counts ≥ 500.”. an unusual cutoff for read counts? What is the rationale and where did you get this recommendation from?
- Supplementary Table S3: There are no flowering-related pathway genes listed in this table. It is only GA related genes.
- Line 299: “Quantitative RNAseq” – What is this? Isn’t RNAseq quantitative?
- Figure 6 caption should mention What are panels A, B, C, & D.

Author Response
Enclosed please find our revised manuscript entitled “Gibberellin Promotes Bolting and Flowering via the Floral Integrators RsFT and RsSOC1-1 in Radish” (Manuscript ID: 773574). We have revised the manuscript in accordance with your and the reviewer’s comments. We have changed main figures Figure 2, 3, 4, 5 and Supplementary Figure S4 and S5. The Figure 2 and 3 exchanged and the Fig. 4A combined with Fig. 5, so the number of main figure is 6. The Supplementary Fig. S4 (new one) is added, so the number of supplementary figure is 5.
Referee’s comments (Reviewer 2)
Thank you very much for your precious reviewing. I have responded to each your comments below.
Major Concerns
- The major conclusions of the manuscript is mainly based on the RNAseq data and the data has many major flaws as follows.
- Q >=30 is the generally accepted norm for RNAseq read quality cutoff. The authors used Q >=20 (I believe it is a typo that authors mention that they used less than (Q <=20) sign?) and read length >=25 bp. Therefore, quality of the reads seems generally low.
® There was a big mistake in Phred quality score Q as your concern. We have revised Q > 20 for obtaining suitable reads in the Result (line: 243) and Q < 20 for discarding aberrant reads in the Method (line: 579) parts.
In this manuscript, we ask for your understanding of the fact that the general parameters are not used. We used phred score based on recent research (Title: An extensive evaluation of read trimming effects on illumine NGS data analysis, 2013, PLoS ONE) that high quality values (e.g. Q > 30) has negative effects on most datasets producing more fragmented assemblies. They suggest the best effects evident for intermediate quality thresholds (Q between 20 and 30).
- Authors have used a single arbitrary time point without compelling evidence to support their choice. Authors chose 6 Days after GA treatment and the justification was (Line 210) “The highest Ft gene expression level was observed at 6 days after GA treatment (Supplementary Figure S1)”. Although, it’s not clear which is that “Ft gene” authors referred to, it is clear (Figure S1) that the transcript levels of many genes tested were increasing with time. How did the authors decide not to monitor the transcript levels after 6 days to decide whether there is another better time point further down
® In this paper, the GA treatment time for RNA-seq and qPCR analyses was determined considering two aspects. First, we refer to recent transcriptome studies that GA responsive flowering gene expressions were examined within 72 hours post GA application (Titles: GA-induced changes in the transcriptome of grapevine, 2015, BMC Genomics; Analysis of transcriptional responses of the infloresence meristems in Jatropha curas following GA treatment, 2018, IJMS). The other is qPCR result of several flowering genes as in Fig. S1, at that time we considered the above the references for the time point. The reason why the reviewer pointed out that the material was not used after 6 days.
® We have added names of flowering time genes, RsMAF2 and RsSOC1, in Fig. S1 (lines: 234).
® We tried to study the flowering mechanism based on the difference of bolting time between two lines according to the GA response, and we thought deeply about the RNA-seq processing time for GA treatment, but as a result, most flowering genes did not respond effectively to GA. We discussed reasons of rare flowering time DEGs in Discussion section (lines 518-524).
- Moreover, the transcript level changes of the major genes are not convincingly GA-dependent and suggest vernalization of 10 days alone can cause such changes (Figures 5 and 6: NH-JS2 vs. NH-JS2_GA). The enriched pathways are all secondary metabolite pathways and not even indirectly related to flowering (Figure 4). If RsFT and RsSOC1 are direct targets of GA then, authors should have seen them as top differentially expressed genes in the GA treated plants. However, RsFT wasn’t significant and RsSOC1 wasn’t any different from untreated plants (Table S4). Authors have to perform a qPCR analysis to find whether FT and SOC levels are differentially altered. Nevertheless, authors qPCR data has consistency issues: Figure S1, RsFLC1 levels go up in both GA treated plant lines compared to untreated plants at 6d and not much different between the two lines. In Figure 6 however, RsFLC levels were slightly down in NH-JS2_GA.
® Independence of GA, vernalization induce a number of flowering time DEGs between two lines as our previous report, however, this transcriptome result did not provide enough flowering time DEGs response to GA. Consequentially, the DEGs response to GA were not enriched in flowering pathways. Therefore, we have changed the Fig. 4B to Supplementary Fig. S4 to guard against meaningless result.
® Although RsFT and RsSOC1 were not detected in DEG analysis due to bioinformatical criteria (Fig. S5, line: 407-411), we figured out that their expression were highly responsible to GA in early bolting line, while not in the late bolting line (Figure 5).
® Two RsFLC transcripts were identified as flowering time DEGs in response to GA with different fold change. RsFLC (TBIU055229) transcript was differentially expressed regardless GA, whereas RsFLC (TBIU004737) transcript was responsible to GA between two lines (Table S4), in addition, the qPCR result of RsFLC (TBIU055229) was not significantly differ between two lines as shown in Fig. S1 (considering the error bars, there was no significance), whereas the expression of RsFLC (TBIU004737) showed significant difference to GA application in Fig. 5.
- Furthermore, FT and SOC1 are downstream floral integrators and the pathway which promoted flowering won’t be reflected by their transcript levels. Cause-effect relationships cannot be established by transcript levels.
Same authors have reported in reference 32 that 35 days of vernalization promotes bolting to 100 % in both lines 30 DAV without the need for GA treatment. This implies that GA mediated promotion of bolting may not be direct, and GA promotes flowering by promoting growth. This is further evident by the fact that even treating the seeds with 10 mM GA and then vernalize for 10 days (also see comment 2 below) could promote flowering almost up to 80% in NH-JS2 (Figure 1B).
® We make a decision that GA can not directly affect transition to flowering because any phenotypical changes occurred without vernalization as shown in Fig. 1B: 0 day (no vernalization), 0, 0.1, and 10 mM GA and in Fig. 3A (new version): 0 days vernalization, +GA. Radish is an obligate vernalization-required crop. Therefore, we investigated the transcript levels of flowering genes in response to GA under marginal vernalization. There are some reports about the role of GA in insufficient vernalization (Title: Gibberellins act downstream of Arabis PERPETUAL FLOWEING1 to accelerate floral induction during vernalization, 2019, Plant Physiol.; The interaction of plant growth regulators and vernalization on the growth and flowering of cauliflower, 2004, Plant Growth Regulation).
If the authors would want to make the claim that GA directly promotes flowering via RsFT and RsSOC1 as they did in the manuscript, they need an RNAseq time course to show that at least FT and SOC1 levels change meaningfully with time in GA dependent manner. Otherwise, authors need to modify the manuscript to admit the above reality.
® We have changed the title, subtitle and main body including “under marginal vernalization” to illuminate the expression levels of RsFT and RsSOC1 in bolting and flowering transition. Despite unsuccessful flowering time DEGs in RNA-seq, the expression levels of RsFT and RsSOC1 showed distinct responsiveness to GA in line with flowering characters (Fig. 5).
- Figure1: Seeds were treated with GA. How persistent (non-degradable) is GA to be responsible for promoting bolting many days after germination? The observed differences among treatments could be simply due to promotion of germination and early growth, by GA and therefore the treated plants responded differentially to 10 d and 20 d vernalization treatments. Figure 3 suggests it is the case. Figure 3 should be Figure 2 to explain that fact. Differences observed in the two lines (NH-JS1 and NH-JS2) is simply inherent response to vernalization treatments. Reference 32 showed that the two lines respond differentially to vernalization treatment.
® We applied GA to seed and soil-grown plants taking a reference (Title: Gibberellins promote flowering of Arabidopsis by activating the LEAFY promoter, 1988, The Plant Cell).
® We have exchanged Fig. 2 and 3 as your suggestion.
® Above mentioned, we have changed title and main body including “under marginal vernalization to avoid direct effect of RsFT and RsSOC1 against GA
- Line 174: Authors claim that “but GA cannot independently induce the floral transition without vernalization”. However, authors did not include appropriate controls in Figure 2 to make that claim. Authors need NH-JS1 and NH-JS2 plants without vernalization and with or without GA treatment. Controls in Figure1 are not relevant for Figure2 (see comment 2 above).
® The Fig. 3A (new version) showed treatment conditions that with and without GA application were taken under 0 days vernalization, therefore 0 days vernalization was triggered.
- Lines 185-186: Authors state that “Therefore, we conclude that GA treatment differentially affects Ft-related gene expression in the two inbred radish lines “. At this point authors haven't shown any data yet. Authors may use “hypothesize” instead of can't “conclude
® I agree with your comment, so I have deleted this sentence.
- Lines 319-320: Based on the RNAseq data authors claim that “This suggests that regulators of GA biosynthesis and signaling are involved in bolting and flowering pathways as activators or repressors in radish.” Authors need genetic evidence to make such strong claims.
® I agree with your comment, so I have deleted this sentence.
- RNAseq data set is not found in the repository (GEO functional genomics data repository of NCBI under accession number SRP176394) and needs to be submitted before publication.
® We have correct the NCBI accession number to GSE125875 (line: 575)
Minor Concerns
- Authors use an unintroduced abbreviation “Ft” throughout the manuscript: Is this Flowering Locus T (FT) or flowering time genes? Please do not use confusing terminology not used generally in the field. In some places, authors use Ft gene (Lines 75, 210) which confuses even more as to whether this is one gene authors talk about.
® I have changed “Ft” to “flowering time” throughout this manuscript for avoiding confused with floral gene FT in accordance with your comment.
- Figure 1B: How is it possible to have more than 100%? Why is there a 120 in the y-axis?
® We have changed the maximum y-axis of Fig. 1B to 100%
- For each Figure / experiment authors need to mention that how many plants were measured and the statistics. e.g. missing in Figure 2B.
® The Fig. 3A (new version) was refered number of plants used the experiment in the legend and the same plants were measured in 3B. The number of seedlings and plants used in phenotyping also described in Method part.
- Plant samples for RNAseq (Lines 208-210): Missing a lot of important information. Information in Fig. S2 should be highlighted here if the samples were treated in a similar way.
® I have added information about the samples that used in qPCR analysis and RNA-seq (line: 231-232; 235-237)
- Supplementary Table S2 is missing.
® We have added Table S2 in Supplementary Materials
- RNAseq DEG cutoff (Lines 235-236): “DEGs were analyzed using the following criteria: log2 (fold-change) ≥ 0.6, false discovery rate (FDR) ≤ 0.01, and read counts ≥ 500.” an unusual cutoff for read counts? What is the rationale and where did you get this recommendation from?
® To avoid false rate the DEG, the Service Company (Seeders Co. in South Korea) uses read counts > 500. There are references that published results using the similar criteria (read count > 1000) by the same company (Virology, 2018, 516: 1-20; BMC Genomics, 2016, 17:474).
- Supplementary Table S3: There are no flowering-related pathway genes listed in this table. It is only GA related genes.
® We already identified flowering genes of radish in a previous study (Ref [32]), so we listed up GA-related genes in this manuscript.
- Line 299: “Quantitative RNAseq” – What is this? Isn’t RNAseq quantitative?
® I have deleted doubleness “quantitative” word as your suggestion
- Figure 6 caption should mention What are panels A, B, C, & D.
® I have specified each panel in Fig. 5 (new one) figure legend.
I hope that the changes we’ have made in the manuscript will meet with your approval and will allay the concerns of the Reviewers.
I look forward to hearing from you at your earliest convenience. Please do not hesitate to contact m again if you have any other queries.
Sincerely yours,
Hye Sun Cho
Plant Systems Engineering Research Center
Korea Research Institute of Bioscience and Biotechnology

Round 2
Reviewer 2 Report
none
Author Response
Thank you for your reviewing this manuscript